# Dental Education during the COVID-19 Pandemic in a German Dental Hospital

**DOI:** 10.3390/ijerph18136905

**Published:** 2021-06-27

**Authors:** Julia Winter, Roland Frankenberger, Frank Günther, Matthias Johannes Roggendorf

**Affiliations:** 1Department of Operative Dentistry, Endodontics, and Pediatric Dentistry, Medical Center for Dentistry, Philipps University Marburg and University Medical Center Giessen and Marburg (Campus Marburg), Georg Voigt Str. 3, 35039 Marburg, Germany; julia.winter@med.uni-marburg.de (J.W.); frankbg@med.uni-marburg.de (R.F.); 2Institute for Medical Microbiology and Hygiene, Marburg University Hospital, Hans-Meerwein-Straße 2, 35043 Marburg, Germany; frank.guenther@staff.uni-marburg.de

**Keywords:** dental education, COVID-19, online learning, patient treatment by students, SARS-CoV-2 Rapid Antigen Test

## Abstract

Due to the SARS-CoV-2 pandemic, dental treatment performed by undergraduate students at the University of Marburg/Germany was immediately stopped in spring 2020 and stepwise reinstalled under a new hygiene concept until full recovery in winter 2020/21. Patient treatment in the student courses was evaluated based on three aspects: (1) Testing of patients with a SARS-CoV-2 Rapid Antigen (SCRA) Test applied by student assistants (SA); (2) Improved hygiene regimen, with separated treatment units, cross-ventilation, pre-operative mouth rinse and rubber dam application wherever possible; (3) Recruitment of patients: 735 patients were pre-registered for the two courses; 384 patients were treated and a total of 699 tests with the SCRA test were performed by SAs. While half of the patients treated in the course were healthy, over 40% of the patients that were pre-registered but not treated in the course revealed a disease being relevant to COVID (*p* < 0.001). 46 patients had concerns to visit the dental hospital due to the increase of COVID incidence levels, 14 persons refused to be tested. The presented concept was suitable to enable patient treatment in the student course during the SARS-CoV-2 pandemic.

## 1. Introduction

The SARS-CoV-2 (COVID-19) pandemic challenged dental education all over the world. The administration of dental schools had to protect the health of students, faculty and patients, keeping track of changing environment and local or national policies, while ensuring continuity of education [1]. A certain part of dental teaching may successfully be converted from the classroom to online webinars [2], however, a substantial part of dental education is traditionally performed “hands-on” with routine chairside teaching [3]. The situation at the dental school in Marburg, Germany was as follows: Instead of the organisation of dental education in study years here education is arranged in semesters, with one study years divided into two semesters. Typically, lectures and courses in the winter semesters run from mid-October to mid-February, the summer semesters from mid-April to mid-July. In the winter semester 2019/20, patient treatment could be regularly completed in the student courses I (7th semester = 4th year) and II (10th semester = 5th year) in preventive and restorative dentistry. Due to the first wave of the SARS-CoV-2 pandemic, both practical and oral board exams had to be postponed from the end of March to the end of June 2020. The summer semester 2020 started with a one-week delay in the end of April 2020, but instead of face-to-face teaching and supervised patient treatment, the semester started with online lectures only. This theoretical part lasted six weeks using the Cisco Webex platform (Cisco Systems Inc., San Jose, CA, USA). The lecturers of the Department of Operative Dentistry, Endodontics and Pediatric Dentistry (OEP) held their online lectures live during the course hours that were otherwise reserved for patient treatment, showing two main advantages:(1)The lectures and seminars for course I and II provided a firm daily structure, which was important in times of the SARS-CoV-2 pandemic to prevent mental illness [4].(2)The usual lecture and seminar sessions for the face-to-face courses were available for additional online lectures. This allowed other departments to offer their lectures in greater blocks, which, in turn, avoided quick changes between two online courses for the students, because the students did not have one Webex room per semester, but the professors and tutors invited the students to their own Webex room.

At the end of the described theory sequence, the students’ level of knowledge was evaluated with a written examination in presence. This also marked the switch from online lectures to chairside teaching. In the summer semester 2020 the two patient treatment courses were almost completely shut down and replaced by treatment simulation using dental dummies. In the treatment course I, students performed mutual treatment including examination, cleaning of teeth and rubber dam exercises. In course II, each student treated one patient. Following the two practical courses mentioned above, patients were treated in the practical training part of the Department of Periodontology from August 10th-28th. Hand instruments and polishing instruments were used for patient treatment, and additional aerosol formation by water-carrying instruments was avoided.

The assessment at the end of the summer semester 2020 was carried out as follows: Dental faculty members were prepared for digital education, teaching videos and scripts were created to offer a theoretical basis for the students. Prior to rearrangement of patient treatment under supervision, a hygiene concept had to be developed to enable students to treat patients on a larger scale than during summer 2020, even during the SARS-CoV-2 pandemic. For the case that the dental school failed in this aspect, untrained dentists would be graduated from the university. In addition, for all other students being in the clinical education section, the lecturers felt obliged to compensate for deficits in terms of practical skills, patient communication, and patient management caused by the low number of patient treatments up to the exam in the next courses.

A general hygiene concept was developed by the Dean of the Dental School in consultation with those responsible at Philipps University of Marburg, administration of the University Hospital Giessen/Marburg (UKGM) as the operator of the dental clinic, and the division of infection control at the UKGM to enable patient treatment under supervision for the students. Patient treatment carried out under supervision in the winter semester 2020/21 was analysed based on patient files. The intention of the present retrospective observation was to investigate the patients’ willingness to participate in the students’ courses and to compare the number of performed treatments before and during the SARS-CoV-2 pandemic.

## 2. Materials and Methods

### 2.1. General Hygienic Measures Due to the SARS-CoV-2 Pandemic

In principle, patient treatment in the student courses during SARS-CoV-2 pandemic is based on three main aspects, which are explained in detail below:(1)Testing of the course patients with a SARS-CoV-2 Rapid Antigen Test (SCRAT).

Prior to any patient treatment in the student course, patients were tested daily by specially trained students using a SCRAT (Roche Diagnostics GmbH, Mannheim, Germany). The waiting and test areas were realised by two tents built on the parking lots, one as waiting area before testing and another area, where patients were tested in four separate areas simultaneously. Up to 17 patients were tested to be eligible for student treatment in the course rooms of the OEP Department in the mornings and afternoons. To avoid large gatherings of patients and corresponding waiting times, patients were appointed at two times in the morning (07:30 and 08:00 a.m.) and in the afternoon (12:30 and 1:00 p.m.), respectively. After a few weeks, it became apparent that the time span between the two test rounds in the morning and in the afternoon resulted in waiting times of around 15 min for the testers. Therefore, the times for the patients to appear at the clinic were corrected to 7:30 and 7:45 in the morning and 12:30 and 12:45 in the afternoon. After SCRAT, patients waited for the test result in Building B of the dental clinic (Figure 1), wearing facemasks and in compliance with the distancing rules. In favour of a negative test result the patients were personally picked up by the attending student and guided to the individual treatment chair.

In case of a positive SCRAT, the patient was led back to the test area where the material was taken for further SCRAT testing. Until the PCR test result was available, the patient was advised to go into quarantine (*n* = 1; positive SCRAT and positive PCR test). A total of 30 student assistants (SA) were trained and employed to test the course patients at each course day during the winter semester 2020/21. These students were already in the clinical training phase of their undergraduate degree in dentistry. Therefore, SAs had a great interest to ensure supervised patient treatment to a greater extent than in the previous summer semester. SAs were recruited from different semesters (from the 7th to the 9th semester) with always 10 SAs having been present simultaneously. One SA welcomed the patients to be tested and handed out the prepared documentation forms for the test and the waiting numbers. A second SA accompanied the patients to be tested on their way to the waiting area. A third SA guided the patient into the next area where the person was tested by one of the four SAs who carried out the SCRAT. After leaving the test area, another SA took over the patient and showed the way to the post-testing waiting area in clinic Building B (Figure 1).

In the test area itself, two more SAs were responsible for documentation, who also called the departments in case of a negative test result and gave permission for the respective patient to be picked up from clinic Building B. Without additional employment of the SAs, testing of the patients would not have been possible, because the human resources (HR) (−20% during the last 5 years due to underfunded budgets) of the dental school are not sufficient to additionally delegate two persons from each department for the testing. Beside SAs, one dentist was always present in the test area for support, feedback on patient communication as well as patient management, and being afterwards available for any patients’ questions. In the morning, this meant that the dentist on duty in the test area had to start work at 7 a.m. instead of 8 a.m., and in the afternoon, having 30–45 min less for lunch. All faculty members were willing to work these overtime hours in exchange for time off to enable the treatment of a maximum of 17 patients per half day in the student course. Testing of additional patients later in the day was not possible due to other lectures and HR of scientific and non-scientific staff.

Due to the immediately initiated purchase of 19 patient simulators, which can be directly mount onto the treatment chair after removing the backrest, treatment at the dental dummy was still possible in case of short-term appointment cancellation by a course patient or limited treatment needs at the respective patient. The practice of practical skills as a hybrid education of patient and simulator training was well-accepted by the students. Especially for the students in the 10th semester, constructive criticism was offered at a high level by the supervising assistants during treatment on the patient simulator to teach the students further refinement of practical skills. Due to the fact that with dummy treatment no patient was present the assessment was easier to communicate. Further the students had the same exercises that allowed a comparison between their treatments. Additionally, in order to avoid a potential effect of the supervising assistant the groups of supervised students were switches in the middle of the semester.

Only those patients for the student course who were treated in the large course rooms of the OEP Department were tested. All other daily patients undergoing treatment in closed operatories were not tested with SCRAT. For the cohort of patients without testing, body temperature was measured in the entrance area of the dental clinic, and they were asked about well-known symptoms of a SARS-CoV-2 infection. When there were no symptoms of illness and the temperature measurement was below 37.5 °C, patients could head to the waiting area of the respective department in Building A. In this way, there were no prolonged encounters of tested and untested patients in the building. All patients with negative test results were treated in the course room and all untested patients without SARS-CoV-2 infection were treated in separate treatment rooms by dentists only. Patients with acute dental pain and an existing SARS-CoV-2 infection were treated outside the test times in a special infection room in Building B. During the treatment of pain patients with an existing SARS-CoV-2 infection, the dental team wore full-protection equipment like the testers in the test area.

### 2.2. Hygiene Concept in the Dental Clinic and in the Course Room

For all lab and patient courses, the dental clinic was reopened at fixed times by the respective supervising course assistants. A total of three entrances equipped with disinfection dispensers were available for this purpose. For contact tracking, students were urged to scan the QR code in the entrance area with their cell phones. When the students arrived at the dental clinic, body temperature was measured, and they were interviewed about possible symptoms regarding a SARS-CoV-2 infection. In the winter semester 2020/21, access to the clinic was only allowed for students wearing both face mask and face shield. During the SARS-CoV-2 pandemic the behaviour of students in the dental clinic was regulated by an operating instruction. The operating instructions and their strict observance were explicitly referred to the course regulations of course I and course II. Repeated disregard of the operating instructions could lead to exclusion from the course.

The changing area for the students of the 8th and the 9th semester moved from the basement of Building A to two rooms in Building B to avoid gatherings in the student changing areas in the basement. All other students continued to change their clothes within their semester bubble in the basement areas at different times, so that longer-term encounters of students from different semesters were avoided. The changing area for the students of the 8th semester and the 9th semester in Building B was clearly demarcated from the patient waiting area after testing.

Before the onset of patient treatment in the winter semester 2020/21, clear double web panels were fitted between the treatment units to separate them. The material counter was protected from possible aerosol transfer between treatment units by acrylic glass panels on both sides. Figure 2 shows the course room of the OEP Department before and after the separation of the treatment units. Their installation in combination with the SCRAT was requested to allow a patient treatment in the students’ courses.

The large windows on both sides of the course room allow a good cross-ventilation (Figure 2). In addition, three air purifiers (2000i, Philips, Hamburg, Germany) were used to purify the air in the course room. Before treatment, patients generally rinsed with a mouth rinse containing chlorhexidine for 30 s [5]. Regardless of the SARS-CoV-2 pandemic, depending on the treatment, the oral cavity is always shielded from the teeth to be treated by placing a rubber dam, if possible.

### 2.3. Patient Recruitment

In contrast to large cities with a medical faculty, students come to Marburg and usually leave Marburg again after completing their courses, but the rest of the population in Marburg county is relatively settled down. Due to this background, many patients in the student course have already been course patients for several years. Typically, the OEP Department can access a pre-registered pool of 200 to 300 adult course patients for Course I and about the same number of patients for Course II each semester. There is also a pool of less reliable course patients and those with low treatment needs. These patients were called centrally by the student counselling service with a fixed appointment before the SARS-CoV-2 pandemic. In addition, there is a pool of paediatric patients for course II. Depending on the number of students, this overall patient pool is mostly treated by the students at the end of a semester. While at the beginning of a semester the typical frame is always that one patient is seen per half day, from the middle of the semester at the latest, the students more often divide the treatment time into two time slots for patient treatment.

With the first lockdown due to the SARS-CoV-2 pandemic, patients’ availability for student treatment was strongly affected. The dental team of the OEP Department was aware that urgent and already started endodontic treatments, which were planned for the student course, had to be continued to keep patients in the dental clinic. After four weeks of lockdown and only emergency treatment in the clinic, dental treatment was restarted and urgent treatments for course patients were carried out in a prioritised manner. In early June, patients having been pre-registered for the summer semester 2020 were offered a check-up appointment and, if necessary, appointments for dental cleaning. For each patient, a treatment of up to one hour was performed by a dentist with 40 years of professional experience, so that there was sufficient time to replace restorations or initiate additional therapy. If necessary, additional appointments were scheduled with the respective patient. Until the start of patient treatment in the student course from early December 2020, patients from the patient pool pre-registered for the winter semester 2020/21 had already been treated, as it was evident that due to the testing of one patient per half-day, it was not possible for the students to treat the whole patient pool.

Evaluation of the patients treated in the student course of the OEP department in the winter semester 2020/21 was as follows: The winter semester 2020/21 started two weeks later than usual on the 2nd of November 2020. First, the students in course I and course II of conservative dentistry were taught all theoretical knowledge in online lectures over 4 weeks. The practical course then started with 39 students in the 7th semester with an introduction week, and from the 7th of December 2020 to the 23rd of February 2021, patients were treated on a total of 34 treatment half-days. In course II, the procedures for patient testing were simulated on the 30th of November 2020 and patients had already been centrally called in for the students for the three following days. Compared to course I, patient treatment in course II was only carried out until the 9th of February 2021, because board exams already began shortly after. Thus, patients were treated on a total of 29 treatment half-days in course II.

### 2.4. Study-Specific Methods

The patient files of all patients registered for course I and course II and for the persons effectively treated in these student courses in the winter semester 2020/21 were selected. As this was a retrospective anonymous data analysis of 735 files, an ethical approval was not required. Data on gender, age, first appointment in the department, treatment in the fourth quarter of 2020 (yes/no), treatment in the first quarter of 2021 (yes/no), medical history, number of treatment appointments in the student course, and type of treatment were entered anonymously into an Excel file. Based on the information of the medical history form recorded in the patient’s file, a classification into the following three health statues was made: healthy, primary diseases not relevant to COVID, primary diseases relevant to COVID. The following diseases were found to be relevant to COVID: hypertension, cardiovascular diseases, chronic obstructive pulmonary disease both chronic and acute liver and kidney disease, shock, diabetes, and cancer [6,7,8,9].

The type of treatment was classified according to check-up, tooth cleaning, filling therapy in the anterior or posterior region with information on the number of one-, two-, three- and four-surface fillings, number of root canals treated and, for course II, number, and material of indirect restorations.

The postulated working hypothesis was that the number of dental treatments in the students’ courses of the OEP Department shows a decrease during the pandemic winter semester 2020/21 compared with the pre-pandemic winter semester 2019/20.

A statistical evaluation was carried out with SPSS 26.0 (IBM Corp., Armonk, NY, USA). The relation of the following variables age, gender, health status and course treatment on each other were calculated using chi square and Fisher’s exact test. Exploratory analysis was used to determine the frequency distribution of the reasons for refusing course treatment. The level of significance was adjusted at α ≤ 0.05.

## 3. Results

The rental cost for the tents amounted to 1,154.30 € per month. Dependent on the outside temperature the electric 15 kWh heater and 10 smaller additional heaters with in total another 15 kWh were running up to 24/7 from November 2020 to February 2021 resulting in costs that could not be calculated exactly. The costs for testing the course patients were about 7.60 € per patient and included the personal protective equipment of the testers, SCRAT, hand disinfectant, wipe disinfection, discard containers for the evaluated tests as well as other materials. 

The SAs were paid 12.12 € or 14.15 € per unit of 45 min depending on their study progress. Based on the number of patients called in, the SAs work 90 min per half day (including preparation, cleaning and changing of the clothes). In the winter semester 2020/21, of the 735 potential student course patients in the OEP department, 384 patients were indeed treated in the two courses (Table 1).

On these 384 patients, a total of 699 tests with the SCRAT were performed by the SAs. For each treatment half-day (63 half-days in total), about 11 tests were carried out for the department. The maximum number of tests for a patient was eight and 200 patients (52.1%) were treated only once in the winter semester 2020/21 in the student course.

While half of the patients treated in the student course were healthy, over 40% of the patients pre-registered but not treated in the course had a disease relevant to COVID (Table 1, *p* < 0.001). In the total group of 735 potential course patients, the healthy patients were on a median 31 years old, the patients with diseases without COVID relevance were on a median 54 years old and the patients with diseases relevant to COVID were on a median 65 years old. The patients were not normally distributed, but health status correlated with the median age (Fisher´s exact test, *p* < 0.001). The median age of the patients in the two courses differed significantly (Pearson’s chi-square test, *p* < 0.001), with mainly younger patients being treated in course I. In the group of patients treated in the course, there was no normal distribution of the males and females concerning the age distribution (Kolmogrov-Smirnov, *p* < 0.001). The possible non-parametric calculation showed significant differences in the age distribution of the treated patients between both genders (Pearson’s chi-square test, *p* = 0.005). The female patients treated were significantly younger (median age: 35 years) compared to the male patients (median age: 49 years).

Regarding the total number of course patients (351 patients) who were regularly pre-registered for the course treatment in winter semester 2020/21, 163 patient files were not handed out for appointments because of two different reasons: either there was no treatment need (85 patients), or there were no treatment capacities in the students’ courses (78 patients). Those patients with no treatment need received instead an in-clinic treatment by dentists during the COVID-19 pandemic in the special service of the department. Potential additional treatment needed was postponed into the summer semester 2021. This service was intended to keep as many patients as possible in the clinic’s recall system during the pandemic. Table 2 shows the reasons for non-participation of patients called for appointments by the students in the winter semester 2020/21.

Eight percent of the 188 patients were treated by an OEP Department dentist, because it was clear on an early stage that the students would only be able to treat one patient per half-day for reasons of personal resources during testing and hygiene. Thirty patients received further treatment from an outside dentist. Forty-six patients had reservations about visiting the dental clinic because of the COVID incidence levels reaching a peak in November 2020 (7-day incidence: 258.1) [10] and 14 persons refused to be tested (Table 2).

Table 3 shows the health status of the treated and pre-registered non-treated patients in the student course in relation to gender and course type. While 56.5% of the patients treated in course I were healthy, 46.9% of the patients treated in course II had a disease with an increased risk of a severe outcome in case of COVID infection (Table 3, *p* < 0.001).

When investigating the reasons for rejection of patients pre-registered for the student courses and the health status, significant differences were computed between the groups (Chi-square test, *p* = 0.012). The reason for the patients’ cancellations due to the COVID situation in health status groups 1 (healthy) and 2 (primary disease not relevant to COVID) revealed 17.5% and 19.6%, respectively. Thus, patients in group 2 slightly predominant mentioned the pandemic to be the relevant factor for their refusal. Patients of group 3 (primary disease relevant to COVID) were even significantly more afraid of the COVID pandemic expressed by almost one third (32.9%) of these patients not willing to attend at the student course during the winter semester 2020/21.

Table 4 displays the treatment range in student course I and course II of the OEP Department with number of services carried out in the winter semester 2019/20 (before any influence of COVID-19) and in the winter semester 2020/21. Besides the total treatments additionally the treatments per student were calculated due to the different semester sizes.

It is obvious that treatments that could be postponed were also performed in the course. Whereas in the summer semester 2020 no patient had more than one appointment in the students’ courses, in the winter semester 2020/21 an average of five dental examinations and four tooth cleanings per student were performed in course I (course II: two dental-check-ups and one tooth cleaning). The number of restorations placed per student during patient treatment was lower in the winter semester 2020/21 (course I and course II in the average eight fillings per student) compared to the previous winter semester (course I in the average 18 restorations per student, course II 19 fillings per student). If the treatment time was still considered in this comparison between restorative therapy in the winter semester 2019/20 and in the winter semester 2020/21 in course I, there was still a reduction in restorative performance per time and per student, but this was now around 30% and not more than 50%. It can be assumed that these 30% of reduction could be at least partially compensated by showing videos on the topic of filling therapy during the online lectures as well as by intensive restorative training on dummies.

## 4. Discussion

Dental education fundamentally differs from medical education [11]. While medical education will increasingly use tele-health formats [12], an essential component of dental student is performed by clinical training is patient treatment under supervision [2]. With the onset of the SARS-CoV-2 pandemic, patient treatment by students was massively scaled down. Since the challenge for dental education is to train competent dentists who are “fit for purpose” after graduation, a concept had to be found to allow patient treatment under supervision on a larger scale than in the summer semester of 2020. With the concept presented here, the educators at the dental school in Marburg, Germany have succeeded in offering patient treatment for students again. Surely this concept had certain expenses, but thanks to the hospital’s purchasing policy, the test could still be offered comparatively inexpensive. The costs for outsourcing patient testing to a heated test area were not considered in the cost breakdown because this can certainly be arranged differently at other dental clinics. In Marburg, this strategy is not due to hygiene and distance rules, but primarily to the limited space available in the dental clinic. An essential factor for the presented test concept was the employment of student assistants (SAs). Due to the loss of many other earning opportunities around gastronomy or exhibitions [13], the SAs were grateful for the work in which, in addition to the testing itself, patient communication and patient management were the focus. Patient communication and behaviour management are important elements of dental work that need to be practised and exercising without patients is associated with great efforts, as in medicine, through role plays with simulation patients in the objective structured clinical examination [14].

Compared to many other students and schoolchildren, dental students have a regular daily routine, and social contacts within their “semester bubble”. Both aspects are important to prevent mental illness during the COVID-19 pandemic [4,15]. It was clear to all students and dental staff that in case of misbehaviour in terms of hygiene and distance regulations, chairside teaching may have to be stopped completely in the event of a COVID-19 outbreak in the dental clinic. Therefore, so far, all persons involved have behaved very carefully in order not to disrupt the presented concept of dental education. There were patients treated in separated treatment rooms without testing who showed COVID-19 infection shortly after dental treatment. However, due to the protective measures, dental staff in Marburg did not become infected by these patients, as evidenced by appropriate testing of the staff. As recommended, the dental staff, as well as dental students, wore during patient treatment a fit tested FFP2, N95 or KN95 respirator, eye protection, a gown or protective clothing and gloves [16,17,18].

The 19 non-separated dental chair units in the course room of the OEP department represented a substantial risk for virus spread [19], therefore the installation of the double web panels between the treatment chairs was important. Many dental procedures such as the use of rotary instruments with water irrigation produce aerosols and droplets that can be contaminated with viruses [20]. During dental treatment, it is almost impossible to avoid large amounts of aerosols and droplets mixed with the patient’s saliva and even blood [21], but their transmission should be blocked [22]. In their review, Peng et al. examined different strategies. Many of the practical strategies for blocking transmission of SARS-CoV-2 infection during dental treatment mentioned by Peng et al. [22] such as patient assessment, hand hygiene, personal protection measures for students, mouth rinsing before dental procedures, rubber dam isolation, and surface disinfection were used in the student course. In addition, the temperature of students was measured at the entrance to the dental clinic. Especially on the cold winter days, the students sometimes had to warm up a little bit like it is described by Erenberk et al. [23] to make a temperature measurement possible.

Studies from the last century already showed effective and efficient bio-aerosol suppression using rubber dams during dental procedures [24,25]. However, the advantages of the rubber dam application must be lectured in the student course and trained on the patient under supervision so that it is routinely used in independent practice after graduation [26]. Despite the SARS-CoV-2 pandemic, treatment under a rubber dam is a basic component of student training in Marburg, which is also relevant to the final exam. In the third year, treatment under rubber dam is first educated and practised for the simulation treatment using a dummy. Then, the students practise it on each other under supervision and afterwards the rubber dam is used in all student courses with patient treatment. The students in Marburg are trained to apply rubber dam even in the case of difficult tooth positions or deeply fractured teeth. If the students declare that they cannot place a rubber dam due to the clinical situation of the teeth, the course lecturers assist help them to place it prior to a final decision to perform the treatment under relative isolation. In a systematic review, Samaranayake et al. [27] investigated the efficacy of bioaerosol reduction procedures in dentistry and concluded that employing combination strategies of rubber dam, with a preoperative antimicrobial mouth rinse, and high-volume evacuators may contain bio-aerosols during dental treatment. Large dental treatment rooms have shown certain effects in terms of bioaerosol distribution [28]. As proposed by several authors, cross ventilation using the opening of windows and air purification devices were shown to be highly effective methods to reduce the risk of airborne transmission of COVID virus containing particles [29,30,31,32]. The course room (326.2 sqm) in contrast is equipped with 20 windows in total, each measuring 4 sqm and thanks to the fact that they can be opened this allows a rapid ventilation of the complete course room. In case of decrease of the outside temperature with no permanent opening of the windows possible there were three mobile air purifiers available that supported the ventilation in combination with shock ventilation.

By testing course patients with SCRAT, the strategy of patient assessment for blocking transmission of an infection during dental treatment patient assessment one step further than described by Peng et al. was taken [22]. The cumulative specificity of the SCRAT used to test the patients of the Dental Clinic in Marburg was reported with 98.5% [33]. Due to the lower sensitivity of Antigen point-of-care tests versus virus detection by real-time RT-PCR, the used test might not have the power to exclude an infection in the early and later stage of COVID-19. Patient testing does not provide complete protection from COVID-19 infection in the students and supervising staff, but testing can support efforts to limit transmission [33]. The routine use of the SCRAT for testing the course patients works as a suitable tool to obtain results in real time. Additionally, the psychological effect of these tests should not be underestimated as the patients had a high feeling of safety. All these procedures allowed reintegration of the patient treatment in the students’ courses which was priceless and resulted in a clear proof-of-concept showing no infections among students and dental staff.

Despite all effort being performed to enable patient treatment in the winter semester 2020/2021, the percentage of patients treated in the two courses was lower compared to pre-pandemic numbers with a reduction by −10.0% (root canal fillings in course I) to −80.3% of treatment per student regarding tooth cleanings depending on the type of treatment and treatment course (only an increase of dental check-ups by +8.7% in course I due to mutual inspection in the students’ course) (Table 4). A decrease of patient treatment performed in patient courses due to less capacities, prioritisation of treatments, and refusal of patients for treatments during the pandemic. The percentage of treatments per student dropped by 50% for indirect restorations, up to 80.3% for tooth cleanings and up to 63.3% for anterior and posterior fillings, respectively. In contrast, the decrease of root canal treatments was only −10%/−32.1% and therefore an indicator for the fact, that urgent treatment was preferred.

It was evident that older patients significantly refused the dental treatment during the pandemic. Regarding the reason for the patients to refuse dental treatment, significant differences were observed. Especially older patients prone to severe COVID infections in case of pre-existing COVID-relevant primary diseases were afraid to participate at the students’ courses expressed by the fact that they refused their participation almost twice if a COVID relevant primary disease was present. Similar findings were reported by González-Olmo et al. [34] who conducted a survey on 961 participants on the visit of dental services before COVID-19 lockdown and termination of the total lockdown. They found a significant association between avoidance behaviour for dental care among respondents who had a high COVID-19 fear and those who were older than 60 years [34]. Indeed, decrease of dental visits during the pandemic can be observed worldwide [35,36,37,38,39]. A study from the University of Giessen mentioned that dental care concepts during pandemics are time-consuming [40]. However, elective dental treatment can be postponed. This prioritisation was also done in the dental clinic in Marburg. Here, the tooth cleanings were postponed and more urgent treatment (fillings, root canal treatment) was carried out. Thus, the prioritisation of dental treatment with holding back patient files with elective treatments was therefore postponed according to the recommendations of existing literature [17,41,42,43].

In the present study, course I patients were significantly younger and logically obtained treatment with less treatment time needed (e.g., clinical inspection, direct restorations, treatment of each other etc.), significantly more patients were treated in that course. In contrast, time-consuming and challenging treatment such as endodontic treatment in molars or indirect restorations were generally performed by more experienced students in course II. Hence, the patients selected for the course II were significantly older. Moreover, their increased age was associated with a higher number of COVID-relevant primary diseases. However, older patients required indirect restorations more frequently so the patient selection for the treatment program in course II subsequently focuses on older patients needing larger restorations and more often predominantly performed as indirect restorations. Regarding the health status significantly more male patients revealed primary diseases. Therefore, it was not surprising that male patients refused dental treatment significantly more often than female patients.

To offer the students the opportunity to obtain a certain treatment level, it was mandatory to establish a hybrid education. Due to the reduced number of patients participating in the student treatment courses and the limited number of patients per half-day, the program was completed by the treatment of simulation models in phantom heads. Thanks to the new dental units special dummies with a click-in mechanism allowed a simple exchange of the back cushion of the dental units against the dummy which enables a sufficient fixation of the dummy patient. This option allowed for a fast switch from patient treatment to dummy treatment in the case of a sudden appointment cancellation. In addition this type of mounting by the click-in mechanism allowed a better fixation of the dummies compared with the strap fixation of the old version of dummies that were partially used in the delayed spring board exam in summer 2020. The main advantage was that the reduced patient treatment could be compensated by the treatment of simulation patients allowing a larger treatment range compared to patient treatment alone. This hybrid education expanded the variability and allowed an increased experience of the education program due to the limitations during the pandemic. A lack of adequate undergraduate dental training due to the pandemic reveals an impact on the clinical competence of students [44] and this may affect the overall preparedness and confidence of graduates [45]. The authors believe that the described concept will play a key role in the process of achieving clinical competence for all dental students currently in the clinical teaching section. As mentioned above, 19 simulation heads were purchased, with benefits for subsequent semesters regarding additional training. It remains to be seen whether or not the implemented hygiene concept was adequate to prevent patients from dropping out from the clinic, however, it can be assumed that the significant efforts reported in this paper will be successful.

## 5. Conclusions

The COVID-19 pandemic was challenging throughout the world. Regarding patient treatment special effort had to be carried out to maintain dental education. The presented hygiene concept allowed an adequate reduction of the risk of virus transmission which is especially present in large course rooms. The implementation of online education and patient simulators allowed for continuing of education in the treatment courses and the board exams in the peak of the SARS-CoV-2 pandemic. Based on the positive experience in the treatment of dummy patients as additional patients in the ongoing pandemic appeared to be a sufficient tool until the establishment of SCRAT in combination with air purifiers was shown to be a concept that was well-accepted by patients, students and dental health care personnel. Nevertheless, all given measures significantly improve the preparedness for further pandemic-like scenarios in the future.

## Figures and Tables

**Figure 1 ijerph-18-06905-f001:**
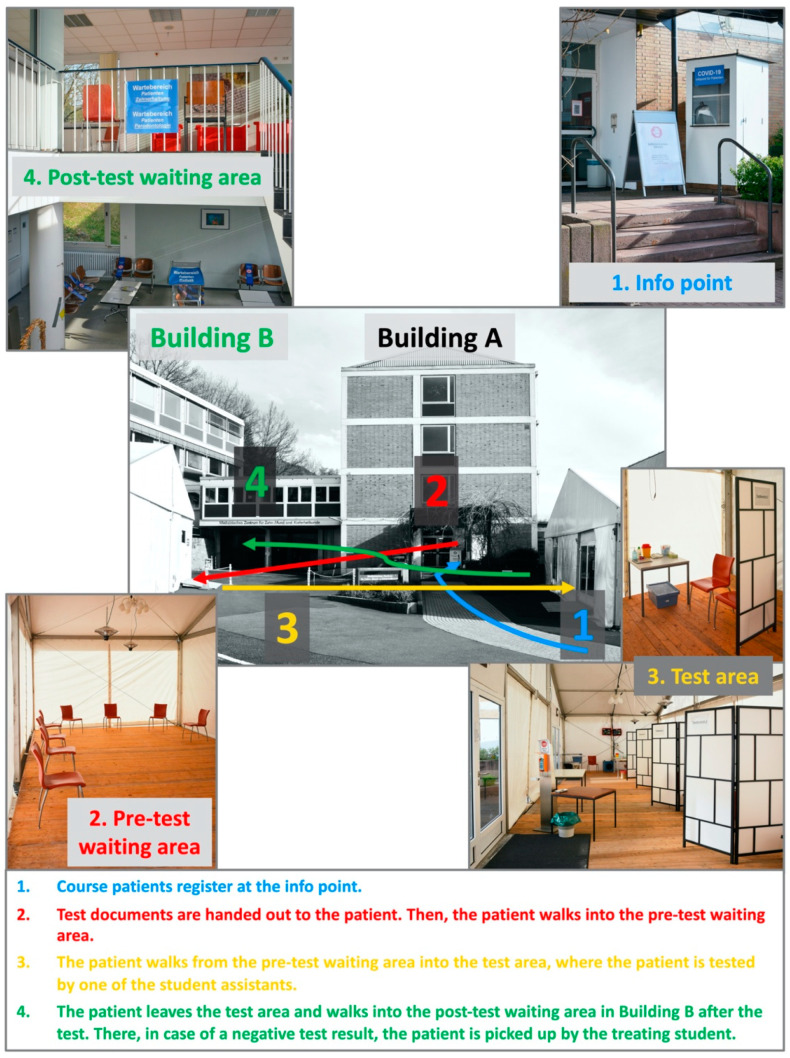
Setup of the patients’ waiting and test area in front of the dental clinic.

**Figure 2 ijerph-18-06905-f002:**
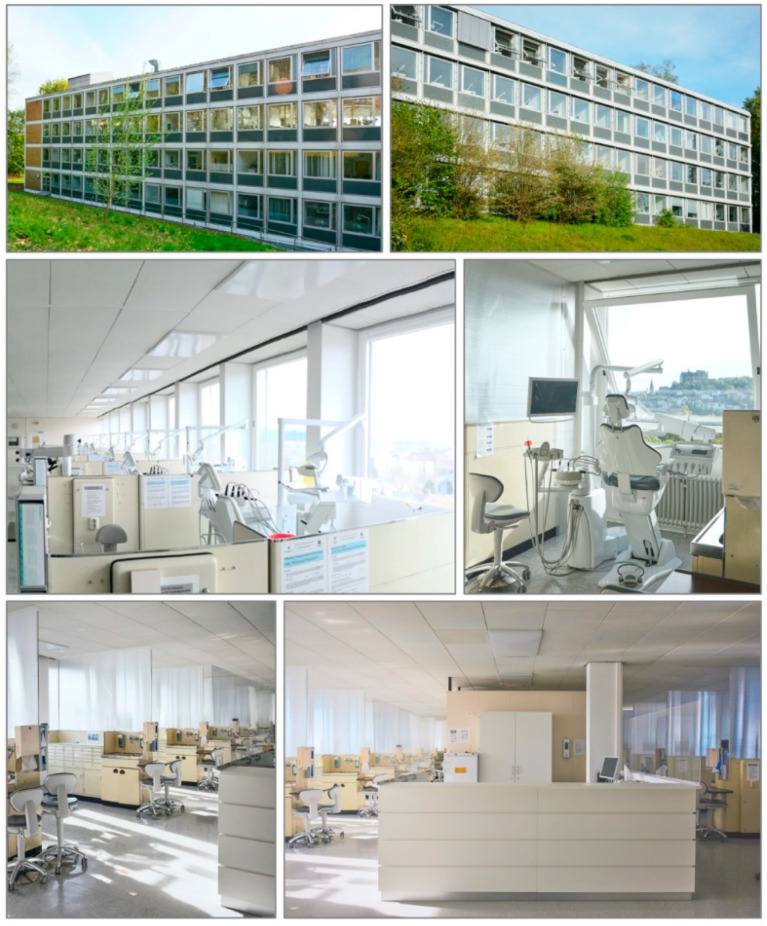
The dental clinic of Marburg (Building A) from the outside (rear/forest view left, City view right; OEP Department on the top floor). The arrangement of the course room of the OEP Department is shown in the middle row (left) with the course room before installation of the separations and new dental units in shown allowing a click-in mounting of the dummies in the dental chairs (2019), the right image in the middle row shows the new dental units with existing separations mandatory for the patient treatment in the students’ courses during the pandemic.

**Table 1 ijerph-18-06905-t001:** Patients treated and not treated in the student courses in relation to gender, median age, health status and course affiliation.

		Patients Treated in the Students’ Courses (*n* = 384)	Patients not Treated in the Students’ Courses (*n* = 351)	Significance Level Two-Sided *p* Value	
				Chi Square by Pearson	Fisher’s Exact Test
male		187 (48.7%)	188 (53.6%)		0.209
female		197 (51.3%)	163 (46.4%)		
median age		43 years	58 years	<0.001	
health status	healthy	193 (50.3%)	119 (33.9%)	<0.001	
	with disease, not COVID relevant	67 (17.4%)	87 (24.8%)		
	with disease, COVID relevant	124 (32.3%)	145 (41.3%)		
course I		271 (70.6%)	190 (54.1%)	<0.001	
course II		113 (29.4%)	161 (45.9%)		

**Table 2 ijerph-18-06905-t002:** Reasons for non-participation of pre-registered patients in the student course.

Reasons for Patients’ Non-Participation in the Students’ Courses	Number of Patients not Participating in the Students’ Courses (*n* = 188)
Patient not available	19 (10.1%)
Patient did not agree appointment	3 (1.6%)
Patient cancelled appointment	5 (2.7%)
Patient was not interested	13 (6.9%)
Time reasons	13 (6.9%)
Health reasons	8 (4.3%)
Weather situation	1 (0.5%)
Further treatment by a dentist of the department	15 (8.0%)
Further treatment in another department	21 (11.2%)
Further treatment by a dentist outside the dental clinic	30 (16.0%)
Patient refused treatment due to COVID situation	46 (24.5%)
Patient reject SCRAT	14 (7.4%)

**Table 3 ijerph-18-06905-t003:** Health status of the treated and pre-registered non-treated patients in the student course in relation to gender and course type.

			Health Status of the Patients Treated in the Students’ Courses			Chi Square by Pearson (Two-Sided *p* Value)
			Healthy	with Disease, not COVID Relevant	with Disease, COVID Relevant	
patients treated in the students’ courses	male (*n* = 187)		99 (52.9%)	19 (10.2%)	69 (36.9%)	0.001
	female (*n* = 197)		94 (47.7%)	48 (24.4%)	55 (27.9%)	
	course I (*n* = 271) first part of the 4th year		153 (56.5%)	47 (17.3%)	71 (26.2%)	<0.001
	course II (*n* = 113) (last part of the 5th year)		40 (35.4%)	29 (25.7%)	53 (46.9%)	
	course I (*n* = 271)	male (*n* = 120)	73 (26.9%)	13 (4.8%)	34 (12.5%)	0.041
		female (*n* = 151)	80 (29.5%)	34 (12.5%)	37 (13.7%)	
	course II (*n* = 113)	male (*n* = 67)	26 (23.0%)	6 (5.3%)	35 (31.0%)	0.013
		female (*n* = 46)	14 (12.4%)	14 (12.4%)	18 (15.9%)	
pre-registered patients for course I or II but no course treatment	male (*n* = 188)		69 (36.7%)	28 (14.9%)	91 (48.4%)	<0.001
	female (*n* = 163)		50 (30.7%)	59 (36.2%)	54 (33.1%)	
	pre-registered for course I but not treated (*n* = 190)		73 (38.4%)	33 (17.4%)	84 (44.2%)	0.002
	pre-registered for course II but not treated (*n* = 161)		46 (28.6%)	54 (33.5%)	61 (37.9%)	
	Pre-registered for course I (*n* = 190)	male (*n* = 105)	42 (40.0%)	9 (8.6%)	54 (51.4%)	0.001
		female (*n* = 85)	31 (36.5%)	24 (28.2%)	30 (35.3%)	
	Pre-registered for course II (*n* = 161)	male (*n* = 83)	27 (32.5%)	19 (22.9%)	37 (44.6%)	0.012

**Table 4 ijerph-18-06905-t004:** Treatments carried out on patients in the winter semester 2019/2020 and 2020/21 in the student courses.

	Winter Semester 2019/20 Number of Treatments on Patients in			Winter Semester 2020/21 Number of Treatments in					
Course	I	II	I + II	I		II		I + II	
Number of students	28	29	57	39		23		62	
Half-days for treatment per student	21.5	19.5		14.0		12.5			
**Treatment on**	**P**	**P**	**P**	**P**	**D**	**P**	**D**	**P**	**D**
Number of dental check-ups (total/per student)	129/4.6	69/2.4	198/3.5	194/5.0	-	55/2.4	-	249/4.0	-
Comparison of number of dental check-ups between ws 19/20 and ws 20/21 (total/treatments per student): course I: + 50.4%/+ 8.7%, course II: −20.3%/no difference									
Number of tooth cleanings (total/per student)	329/11.8	207/7.1	536/9.4	176/4.5	-	32/1.4	-	208/3.4	-
Comparison of number of tooth cleanings between ws 19/20 and ws 20/21 (total/treatments per student): course I: −46.5%/−61.9%, course II: −84.5%/−80.3%									
Number of fillings in anterior teeth (total/per student)									
one-surface	111	109	220	64	-	46	-	110	-
two-surface	41	68	109	20	-	16	-	36	-
three-surface	31	30	61	22	-	3	-	25	-
four-surface	10	22	32	6	23	2	31	8	54
total number	193/6.9	229/7.9	422/7.4	112/2.9	23/0.6	67/2.9	31/1.3	179/2.9	54/0.9
Comparison of number of anterior fillings between ws 2019/20 and ws 2020/21 (total/treatments per student): course I: −42.0%/−58.0%, course II: −70.7%/−63.3%									
Number of fillings in posterior teeth (total/per student)									
one-surface	258	211	469	168	-	72	-	240	-
two-surface	65	96	162	49	-	49	18	98	18
three-surface	5	32	36	9	32	8	19	17	51
four-surface	2	3	5	2	-	2	-	4	-
total number	330/11.8	342/11.8	672/11.8	228/5.8	32/0.8	131/5.7	37/1.6	359/5.8	69/1.1
Comparison of number of posterior fillings between ws 2019/20 and ws 2020/21 (total/treatments per student): course I: −30.9%/−50.8%, course II: −61.7%/−51.7%									
Number of root canal treaments/fillings (total/per student)	27/1.0	90/3.1	117/2.1	31/0.8	36/0.9	48/2.1	12/0.5	79/1.3	48/0.8
Comparison of number root canal treatments/fillings between ws 19/20 and ws 20/21 (total/treatments per student): course I: +14.8%/−10.0%, course II: −46.7%/−32.3%									
Number of indirect restorations * (total/per student)		57/2.0				24/1.0			
Comparison of number of indirect restorations between ws 19/20 and ws 20/21 (total/treatments per student): course II: −57.9%/−50.0%									

Legend: P = patient, D = dummy, ws = winter semester, * only course II.

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
