# Peer review of "Dental Education during the COVID-19 Pandemic in a German Dental Hospital"

_ijerph, 2021, doi:10.3390/ijerph18136905_

Round 1

Reviewer 1 Report

I read the manuscript but some major concerns are distubing me. Which kind of study design does your manuscript apply? Please state the precise study design (do not say just retrospective). What is the purpose of your study? What does it add to our knowledge and how could it improve clinical management of patients? In a scientific article it is a common thing to avoid first person sentences, please corect through the manuscript. I believe that the methods do not follow a conventional and ordered scheme. Why did you apply tests like Kruskal-Wallis if you only have two groups. I am suggesting a major revision to allow you to reply to my comments

Author Response

Dear Reviewer 1,

thank you for your valuable comments, that were very helpful for the modification of the manuscript. Please find the attached word file.

Reviewer 2 Report

Dear authors,

thank you for the opportunity to review your manuscript entitled “Dental education during the COVID-19 pandemic in a German Dental Hospital”. Overall, it is very well written and thoroughly understandable. Nevertheless, the majority of the article is narrative and the key question / hypothesis / aim of your study is not pointed out. Also, some other minor aspects should be considered before publication.

  • Lines 46 and 111/112: During the paper, sometimes the years of undergraduate training and sometimes semesters are used. Especially for non-german readers, I would prefer the consistent use of years.
  • Lines 55-59: Did the students in course I only treat dummy patients and those in course II only one real patient or had the students in course II also additional training on dummy patients? Please specify lines 57-59 with respect to real/dummy patients.
  • Lines 78-81: What is the key question / hypothesis of your study? Please specify.
  • Lines 82-230: The majority of the “Materials and Methods”-Section is narrative and describes the general list of measures applied due to the COVID-19 pandemic. I would recommend to focus on Material and Methods in this section as you did in line 230 and following. Perhaps the section can be divided into “general hygienic measures due to the COVID-19 pandemic” and “study-specific methods” or something else.
  • Line 132: Perhaps I have overseen it, but what does “HR” mean?
  • Lines 138-140: Please specify the advantages of patient simulators regarding constructive criticism of the supervising assistants. During my undergraduate teaching, criticism was more or less constructive depending on the character of the supervising assistant ;)
  • Lines 180-183: I guess the information of new dental units in 2019 is unnecessary. Also, the pictures in the last row miss a description. In my opinion, one of the two pictures in the first row, the left picture in the middle row and the left picture in the last row are sufficient.
  • Lines 253-258: Please provide a sum of additional costs (tests and staff), as you write later in the discussion section, that the heating tents on the parking space were not mentioned in the total costs due to the missing space in the buildings.
  • Table 1: The second line (male) has not to be bold typed.
  • Lines 264-265: You write that 11 SCRAT tests were carried out per half-day. In the introduction section, up to 29 patients were mentioned. Please describe the large variety (mean and standard deviation of tests per half-day).
  • Line 283 and Table 2: When 351 eligible patients were not treated in the courses due to several reasons, why are 163 patients mentioned in line 283 and 188 in Table 2? Please add the 163 patients to Table 2 (no treatment need n=85, no capacity in students’ course n=78). What happened to the 78 patients who exceeded the capacity of the students’ courses?
  • Table 3: Please have a closer look at Table 3: some text is bold, which is not bold typed in the corresponding lines and columns. Additionally, column four (“healthy”) has a formatting error – please adjust the width of the column.
  • Table 4: As in the other Tables, some text is bold, which is not bold in corresponding lines and columns. Please adjust the height of “P” and “D” in the third line. The comparison of anterior and posterior fillings between winter semester 2019/20 and 2020/21 is very uncomfortable for the reader: please provide (if possible from the patients records) the distribution of one- to four-surface fillings and for winter semester 2020/21 the sum of “P” and “D” fillings. Additionally, p-values are missing in the Table, while in the discussion section (line 423), significant findings were proposed.
  • Line 434: I guess you mean “decrease” instead of “decreased”?
  • Line 464: The same for “will” instead of “well”?
  • Discussion section: A corresponding paper published 2021 in the IJERPH of your neighbor clinic could be additionally mentioned in the discussion section (Schlenz et al. 2021: Perspectives from Dentists, Dental Assistants, Students, and Patients on Dental Care Adapted to the COVID-19 Pandemic: A Cross-Sectional Survey).
  • Conclusion: All in all, you have a proper results section but the conclusion section is quite short and general. Please specify your conclusions with regard to the key question of your study.

Author Response

Dear Reviewer,

thank you for your valuable comments. 

Please find the point by point modifications in the attached word file.

Reviewer 3 Report

After having read the scientific article received electronically, I sent the following comments:
1. The manuscript is well structured in a logical and orderly way
2. The methodology is clear
3. The results are logical
4. Discussion can improve
5. The conclusion must be short and precise
6 Improve the bibliography (updated)

Author Response

(The authors gave the same response as above.)

Round 2

Reviewer 1 Report

The manuscript appears improved and better organized.